# Assessing the Accuracy of Underwater Photogrammetry for Archaeology: A Comparison of Structure from Motion Photogrammetry and Real Time Kinematic Survey at the East Key Construction Wreck

**Anne E. Wright [1],*** , **David L. Conlin [1]** and **Steven M. Shope [2]**

[1]  National Park Service Submerged Resources Center, Lakewood, CO 80228, USA; dave_conlin@nps.gov
[2]  Sandia Research Corporation, Mesa, AZ 85207, USA; sshope@sandiaresearch.com
*   Correspondence: anne_wright@nps.gov

**Abstract:** The National Park Service (NPS) Submerged Resources Center (SRC) documented the East Key Construction Wreck in Dry Tortugas National Park using Structure from Motion photogrammetry, traditional archaeological hand mapping, and real time kinematic GPS (Global Positioning System) survey to test the accuracy of and establish a baseline "worst case scenario" for 3D models created with NPS SRC's tri-camera photogrammetry system, SeaArray. The data sets were compared using statistical analysis to determine accuracy and precision. Additionally, the team evaluated the amount of time and resources necessary to produce an acceptably accurate photogrammetry model that can be used for a variety of archaeological functions, including site monitoring and interpretation. Through statistical analysis, the team determined that, in the worst case scenario, in its current iteration, photogrammetry models created with SeaArray have a margin of error of 5.29 cm at a site over 84 m in length and 65 m in width. This paper discusses the design of the survey, acquisition and processing of data, analysis, issues encountered, and plans to improve the accuracy of the SeaArray photogrammetry system.

**Keywords:** Structure from Motion photogrammetry; multi-image photogrammetry; 3D modeling; RTK; underwater archaeology; cultural resource management; National Parks

## 1. Introduction

In recent years, the use of photogrammetry to record submerged archaeological sites has developed into a standard documentation tool [1–13]. Photogrammetry allows archaeologists to record information in great detail, while leaving submerged heritage in situ for future study, and it allows the non-diving public to experience underwater cultural heritage without needing to physically visit the site. This mitigates potential disturbances caused by human interference and provides experiences for those without the skills or means to visit underwater cultural heritage sites in person. In archaeological practice, 3D models give archaeologists the ability to extract data from sites and artifacts as new questions arise, without the need to make repeat visits to a physical location. It also decreases the need for expensive and time-consuming artifact conservation processes and allows archaeologists to digitally "preserve" and catalogue artifacts and sites, as well as share data with other researchers.

The techniques, hardware, and software used by heritage professionals to collect and process photogrammetric data are changing rapidly, allowing for greater accuracy, reliability, and detail.

Faced with the vast management responsibility for more than 1.4 million hectares of submerged lands, the National Park Service Submerged Resources Center (NPS SRC) and its partner organization, Marine Imaging Technologies (MiTech), developed a multi-camera, diver-operated photogrammetry platform called SeaArray that allows the user to capture hectares of seafloor in a single photogrammetry model. While the models created by the SeaArray system are compelling in their detail, NPS SRC wanted a clear understanding of how closely they reflected the spatial patterning and archaeological reality of the sites themselves, as the models are of limited scientific use if it is unknown how accurate the data are and what limitations there are in documentation.

NPS SRC documented the East Key Construction Wreck in Dry Tortugas National Park using Structure from Motion (SfM) photogrammetry, traditional hand mapping, and Real Time Kinematic (RTK) survey to test and establish a baseline "worst case scenario" of accuracy for 3D models created with SeaArray, as well as to evaluate the amount of time and resources necessary to capture an acceptably accurate photogrammetry model of a submerged archaeological site. The resulting photogrammetry model had a specific issue known as the "bowling effect," a scooping or bowling in the center of the model. This error was caused by a lack of camera calibration coupled with homogenous underwater terrain. Ultimately, the team found through statistical analysis that, in the worst case, this photogrammetry model created with SeaArray has a margin of error of 5.29 cm.

Underwater archaeologists use photogrammetry to document submerged archaeological sites three dimensionally. There are two different types of photogrammetry utilized by archaeologists: "Structure from Motion" photogrammetry (abbreviated as "SfM" and commonly referred to as "Multi-Image" photogrammetry) and "Stereo-Vision" photogrammetry.

Structure from Motion photogrammetry is the type most commonly used by underwater archaeologists and is the focus of this paper. SfM creates a 3D model from overlapping images by comparing large data sets to identify matching features shared by images. Once matching features within the images have been identified, a three-dimensional structure of the objects or landscapes is reconstructed via complex calculations of camera lens optics and, via that, camera position relative to the objects photographed.

Underwater archaeologists have readily adopted SfM photogrammetry as a site-documentation tool [1–14], but relatively few studies have been done to assess the overall accuracy of the resulting models, particularly on underwater archaeological sites. An initial study conducted by Balletti et al. [2] showed promise, having found accuracy to 6 cm over a site of up to 36 m long by comparing photogrammetry to RTK data. A separate study by Skarlatos et al. [9], conducted using trilateration, suggests that photogrammetry models may be up to three times more accurate than traditional trilateration measurements taken in 2D.

Parks Canada has offered perhaps the most robust test of accuracy to date using underwater photogrammetry to record the site of HMS *Erebus* [3]. After creating a model of the site depicting a 32 by 8.8 m area in Agisoft Photoscan consisting of over 125 million points, the Parks Canada team georectified the model using multibeam echo sounder (MBES) data and compared the model with eighteen reference points from digital surface model (DSM) data. Ultimately, Parks Canada found differences ranging from 0.4 cm and 2 cm between DSM and MBES data but confirmed problematic depth measurements in the DSM data with discrepancies between 1.8 cm and 35 cm. Agisoft Metashape Professional's internal tool estimated the measurement precision of the model to be approximately 0.5 cm on average.

Several studies have been conducted assessing the accuracy of terrestrial photogrammetry survey using aerial drones and compared to RTK survey. Using ten ground control points, Barry and Coakley [15] found that aerial drone photogrammetry survey was reliable within 41 mm horizontally and 68 mm vertically with a 1 cm ground sample distance. Uysal et al. [16] found a digital elevation model created with aerial drone photogrammetry to be accurate to 6.62 cm from an altitude of 60 m using 30 ground control points. These studies are the most comparable to the research presented here,

although they were both conducted in a terrestrial setting. To the authors' knowledge, Balletti et al. [2] offers the only comparable study conducted underwater using RTK to date.

Due to the ability of SeaArray to document large swaths of seafloor in one SfM model, NPS SRC wished to test the accuracy of photogrammetry models of large submerged archaeological sites, such as the East Key Construction Wreck. In addition to its size, this site was selected because it is in shallow water with proximity to land that makes it possible to conduct RTK survey and it has readily distinguishable features that are not subject to movement due to wave or current action.

### Historical Context

Dry Tortugas National Park is located 109 km west of Key West, at the furthermost tip of the system of low reefs and islands that comprise the Florida Keys. The park consists of seven small islands within a management area of 259 square kilometers [17]. The Dry Tortugas, named for the lack of fresh water and its abundance of sea turtles, sit at the edge of the major route of maritime ingress and egress for the Gulf of Mexico—the narrow Straits of Florida—which separate the southernmost extent of the Florida Keys from the northern shore of the island of Cuba. At the center of the park lies the huge "Third System" structure of Fort Jefferson, built in the first half of the nineteenth century. The construction of a massive fort in this isolated and inhospitable area is a testament to the strategic importance of the nearby harbors. The proximity of the Dry Tortugas to the Straits of Florida was a factor in their regional political and military importance but was (and is) also a factor in their threat to passing maritime commerce. "Any ships traveling the more than 1200 miles of the United States Gulf coastline will pass close to the Tortugas. The Dry Tortugas pose a serious navigational hazard and have been the site of hundreds of marine casualties [17]."

One such casualty was a 19th century sailing vessel that wrecked while carrying building materials for the construction of the fort. Known as the East Key Construction Wreck, the remains of this vessel are located in approximately three meters of water, 8 km east of Fort Jefferson, as illustrated in Figure 1. The most prominent features of the site are the remains of its cargo: graywacke (a hard, metamorphic sandstone) paving slabs, graywacke flagstones, and hundreds of rounded cement barrels (as seen in Figure 2), which formed when the wooden barrels of dry-packed cement hardened after the ship sank. In 1935, Fort Jefferson and the surrounding waters became a National Monument and a part of the United States' National Park System. In 1992, the National Monument was elevated to a National Park—the highest designation for a unit of the National Park System.

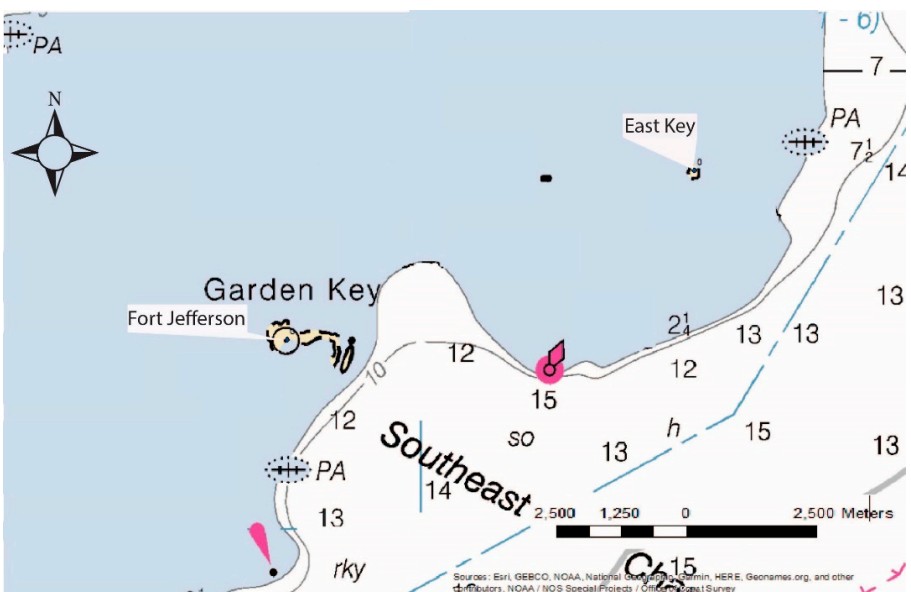

**Figure 1.** Map showing location of East Key Construction Wreck relative to Garden Key and Fort Jefferson.

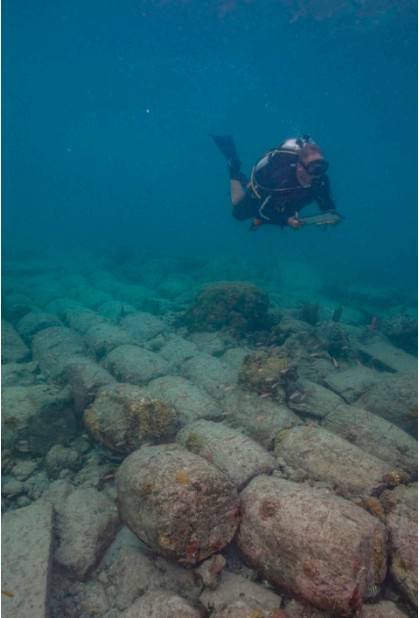

**Figure 2.** An National Park Service (NPS) archaeologist swims over cement barrels on the East Key Construction Wreck.

## 2. Materials and Methods

### 2.1. Real Time Kinematic (RTK) Survey and Measurements

Real time kinematic (RTK) GPS surveying is a method used to enhance the accuracy of GPS measurements. Corrective factors known as differential corrections are applied to the satellite measurements in real time from a second receiver known as a base or reference station occupying a known location on the earth. In practice, atmospheric distortion, multipathing of radio signals, and instrument errors in things, like the GPS satellite clock, introduce distortions in one or more of the satellite signals in the constellation of GPS satellites that is used to generate a position fix for a surveyor. These cumulative errors reduce the accuracy of a GPS position to approximately one meter. Applying real-time, differential corrections from a base station occupying a known terrestrial location can reduce these errors to a few centimeters.

RTK survey requires a reference or base station receiver to occupy a location, commonly referred to as a survey control point, that has an already known position. The differences between the position reported for this survey control point by the receiver and the known position are broadcast to a second receiver or "rover" to correct cumulative errors in the rover's observations. Utilizing RTK techniques and equipment, the NPS archeology team was able to reduce the uncertainty in observed positions from a few meters to a few centimeters (or less) in 3 dimensions—this represents the current state of the art for accurate GPS measurements in a remote, offshore, marine environment and provides the best data available for true real world locations of points within the East Key wreck site.

NPS archeologists established a network of two primary datums and eleven sub-datums (hereafter referred to as markers) for a total of thirteen control points spread out within the East Key shipwreck site, indicated in red in Figure 3. The points were labeled A through M with points A and B set as permanent brass datums at the ends of the site and the other eleven markers denoted by temporary orange plastic stakes. These markers were used to test spatial dimensions of the SfM model and also to add spatial control. The permanent (A and B) datums were set on the outer margins of the site at the north and south ends, while the other markers were set throughout the site and were not set up in a grid or geometric pattern.

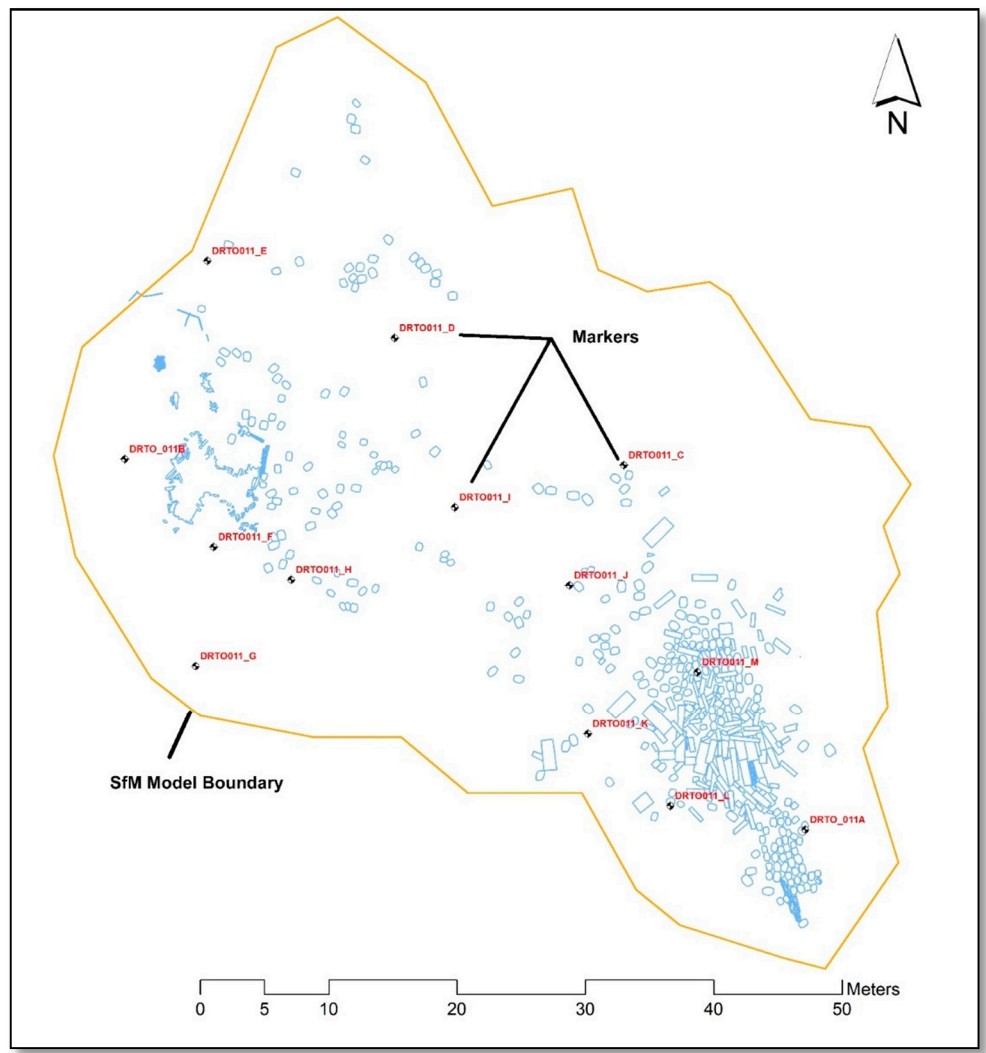

**Figure 3.** Site plan showing Structure from Motion (SfM) model boundary, Real Time Kinematic (RTK) markers, and NPS hand-mapped site plan from 1990.

RTK observations were taken with the rover receiver set up on a tripod over the datum and markers on the wreck site, as shown in Figure 4a. A base station antenna/receiver was set up over a known control point at Fort Jefferson (as seen in Figure 4b) which generated correctional information to broadcast via a Trimble TDL 450 UHF radio to the rover on the site. The equipment used for the base was a Trimble NetR9 receiver connected to a Zephyr 3 GNSS (Global Navigation Satellite System) antenna. Rover operations at the wreck site used a Trimble TDL 450 UHF radio mounted in the survey/dive boat operating as a radio repeater that rebroadcast differential corrections from the base station to a Trimble R8 model 3 GNSS receiver equipped with a short-range internal UHF radio. A weighted 2 m, Hixon Mfg. aluminum tripod was set up and centered on a selected point, and then aluminum extension poles were added until the top extension pole was less than 1 m below the surface of the water. Additional stability was added to the setup by attaching guy lines to this top extension pole and anchored to a solid feature on the seafloor. This minimized excessive movement of the GNSS receiver caused by the effect of wave action on the tripod/extension pole setup. Vertical control for the tripod was via reference to a suite of traditional surveyor's bubble levels attached to the tripod and extension poles. Once the tripod was set up, leveled, and secured, the Trimble R8 GNSS receiver was mounted on top of the last 1 m extension pole by the surveyor in the boat and handed off to the archeologist, who then mounted the last extension pole on the top of the tripod/extension poles,

as shown in Figure 4c. A series of positional measurements were then taken and averaged to provide centimeter accurate positions for each point on the wreck site.

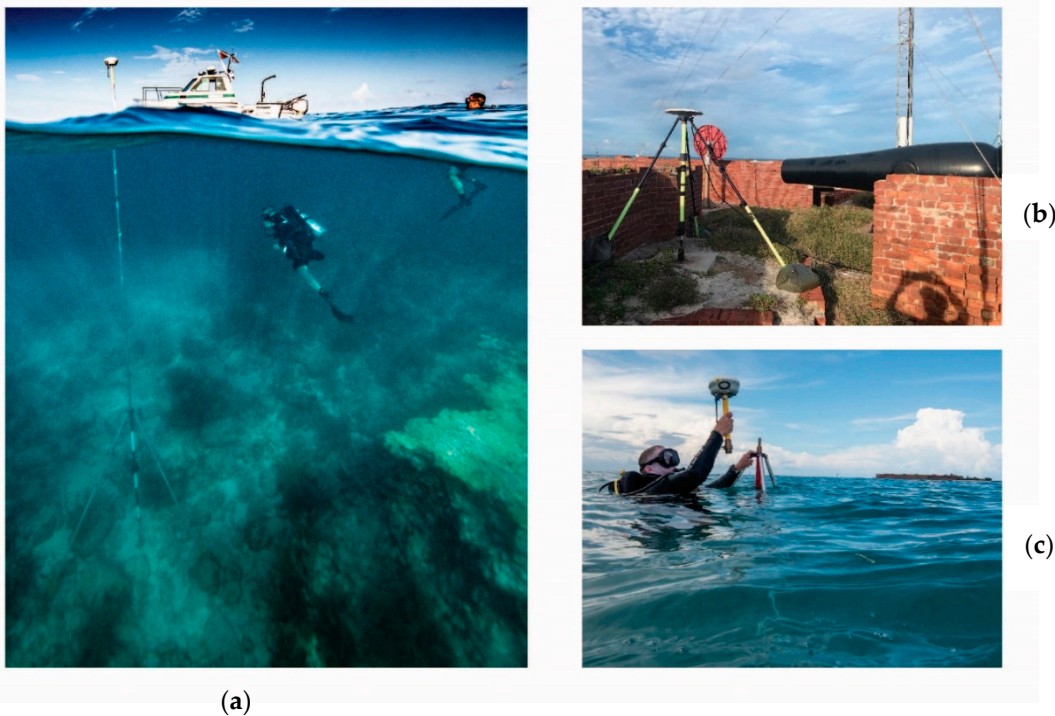

**Figure 4.** (**a**) An NPS archaeologist inspects the tripod from below. (**b**) GPS base station at Fort Jefferson. (**c**) An NPS diver mounts the GPS receiver to the tripod.

## 2.2. Hand Measurement Methodology

In addition to setting up and taking GPS location measurements for a system of datums and markers, the NPS archeology team made direct measurements by hand of prominent features and artifacts on the East Key Construction Wreck site to compare to the same measurements taken in the photogrammetry model. This was done using traditional fiberglass tape measures and recording results on underwater slates with waterproof paper. Measurements were taken to the nearest centimeter.

## 2.3. SeaArray Methodology

To meet the NPS mandate of resource stewardship and protection of submerged sites, the National Park Service Submerged Resources Center, in partnership with Marine Imaging Technologies (MiTech), developed a multi-camera photogrammetry platform named SeaArray, shown in Figure 5. SeaArray is a self-propelled diver-operated three camera array for high resolution image capture that has expanded the boundaries of underwater photogrammetry capabilities from areas of meters to areas of hectares.

The initial impetus for the development of SeaArray was to maximize accuracy with underwater 3D models utilizing a multiple camera acquisition strategy and to increase the area covered during data collection while maintaining high-resolution visualizations, all within the restrictions of time spent underwater and the diver's available gas supply. The multi-camera system has exceeded initial expectations and has been used to survey large 250 m by 125 m areas with pixel-level clarity. With over 340,000 images generated to date, the SeaArray averages 97% image alignment using extremely large data sets and is capable of using nearly 30,000 images for a single 3D visualization.[1]

---

[1]  A high percentage of aligned photos creates models with higher accuracy and larger amounts of detail.

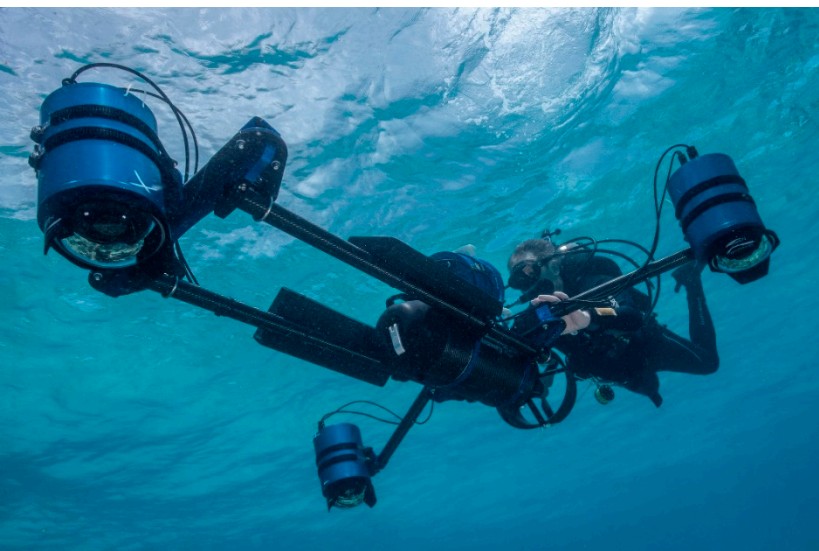

**Figure 5.** A diver operates SeaArray.

SeaArray is comprised of three 45.7 MP Nikon Z7 mirrorless cameras fitted with 14 mm 2.8 Rokinon lenses housed in Marine Imaging Technologies' custom underwater camera housings and camera controller. The imaging system is engineered on a modular, carbon fiber frame with pivoting arms built around a SubGravity/Bonex diver propulsion vehicle (DPV). The system is capable of capturing 10,800 images per hour. The camera control housing includes real-time HDMI (high definition multimedia interface) camera monitoring and switching, ISO control, shutter control, independent still image capture, and a vacuum leak detection system. The Marine Imaging Arduino based software operates the camera controls and shutter releases simultaneously with capabilities for sequential or simultaneous image capturing with a customizable capture rate, synchronized camera settings, and a battery life indicator.

The design of SeaArray as a multiple camera array requires the issue of overlap to be addressed on two levels—individual camera and lane overlap. The "primary" camera for navigation is the central camera in the three-camera geometry. The overall goal in image acquisition is to achieve the Agisoft Metashape recommended 60% side image overlap and 80% forward image overlap. To accomplish this on the individual camera level, overall coverage is largely determined by reciprocal lane spacing that contains 60% side overlap, while the adjustable automated shutter release ensures 80% forward overlap through speed of travel and a typical data collection rate of 1 frame per second. Collectively, the three-camera alignment often offers differing overlap on a single object modeled. During a reciprocal transect, the central camera may be collecting imagery at a 60% overlap, but the inboard camera may be at 90% overlap, while the outboard camera may be closer to 30%. Regarding lane overlap, which consists of a collection of three simultaneously generated images, the goal is again to achieve 60% side overlap and 80% forward overlap. The benefit of SeaArray is that multiple cameras within each lane create additional overlap, which in turn creates more data points for the software to generate. This results in a higher number of tie points within the model.

Underwater conditions are consistently variable from one site to the next. Water visibility, available light, and contour or relief of the object being modeled are factors when considering the distance to the subject or height from the seafloor that the diver must maintain while operating SeaArray. Since the operational goal of the system is to gather as dense of a data cloud as possible, NPS SRC typically attempts to operate 2–3 m from the seafloor or subject, even in good water visibility. This allows the team to not simply map large areas of seafloor but to do so with a saturation of data points that enables the end model to be visualized in great detail.

Advancements in modeling algorithms and the ability of Agisoft MetaShape to generate point locations and measured distances with very little optical testing or calculations was a known benefit

when the development of SeaArray began. The intent was to build an imaging system that was field ready that required little optical engineering or testing that could deliver a highly accurate and robust point cloud of underwater data. The team relies on the software to interpret the required optical corrections and angles (with the exception of camera calibration).

NPS SRC and Marine Imaging Technologies, in partnership with the Department of Design Studies at the University of Wisconsin, developed custom workflows to generate models using Agisoft Metshape Professional (currently version 1.6.3) specific to field project resources, processing availability and time, and end model visualization needs.

For the East Key Construction Wreck 3D model, the team created the model by aligning 100% of the 12,942 images collected from SeaArray's three cameras in Agisoft Metashape. This initial photo alignment process resulted in a sparse cloud containing over 14.5 million tie points, which the team reprocessed by using the gradual selection and camera optimization tools to clean the model of spurious outlying points.

This initial resulting sparse point cloud model fell victim to what is known as the "bowling effect." The bowling effect, which causes a scooping or bowling in the center of a model, can sometimes occur in 3D models rendered from largely homogenous terrain, like the sea floor at the East Key Construction Wreck, when they are generated without proper camera calibration. The seafloor of the East Key Construction Wreck is made up of low-lying artifacts, patchy coral, and sand, all of a similar color palette and height off of the seafloor. The Agisosft Metashape User Manual Version 1.6 [18] suggests that lens calibration can be skipped in common workflows, but may be useful if alignment results are unstable [18]. The issue of lens calibration on the SeaArray system had been discussed and attempted by the authors. The area of optical coverage of each individual camera within the SeaArray system, resulting in the 14 mm (115.7 degree field of view) combined with a fixed focal distance, made SeaArray camera calibration difficult in water. SeaArray camera calibration was first attempted with an initial 2 m × 1 m calibration grid, but it was not large enough to fill the frame while maintaining a standard operational distance of 2 m–3 m. A larger calibration grid has been designed for future calibration of SeaArray cameras.

Due to these difficulties, NPS SRC purposely tested SeaArray without calibrating the cameras in order to test the system at its most basic level, or its "worst case scenario." If the bowling effect occurs in a model created without proper camera calibration, Agisoft recommends correcting this distortion through a camera optimization procedure based on ground control points or camera coordinates after the photo alignment process. During this process, Agisoft adjusts the 3D model based on geodetic parameters introduced by the ground control point data [19]. The team corrected the bowling effect using this methodology, but this "corrected" version of the model is not discussed here, as the primary goal for the study was to test the difference between RTK data and an uncorrected, uncalibrated SeaArray photogrammetry model.

Next, the team generated a 1.2-billion-point dense cloud from the sparse cloud in Agisoft Metashape Professional. The model creation process within Metashape Professional was stopped at the dense cloud step, without continuing to a mesh or textured model, as a high-quality dense cloud is a much more accurate method of visualizing sites. At this point, the software is working only with actually observed data and has not yet interpolated any points, as occurs in the creation of a mesh or textured model. The dense cloud model was exported into Viscore software for data analysis.

Viscore is a point-based visual analytics environment created for the purpose of conducting analysis on large point-cloud data sets by computer scientists of the Cultural Heritage Engineering Initiative (CHEI) at the University of California San Diego. The software allows for quick access and analysis of data [20]. A quicker response time when loading large data sets for model manipulation makes Viscore advantageous over Agisoft when conducting point-cloud data analysis.

Once each version of the model was imported into Viscore, the team scaled the dense clouds based upon three, one-meter scaled targets placed on the seafloor during the data collection process. The position of the markers was then set within Viscore. Using a measurement tool within the software,

the team measured distances between the markers and recorded them in a Microsoft Excel spreadsheet. The Viscore measurement process is shown in Figure 6.

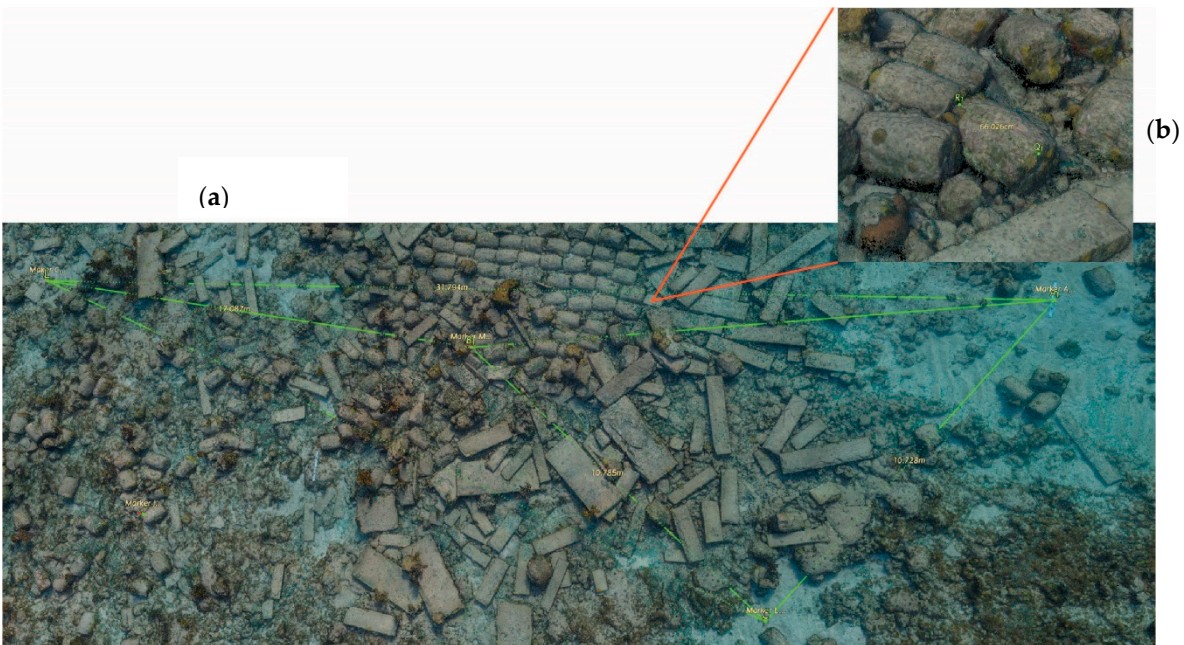

**Figure 6.** (**a**) Measurement of distances between markers in Viscore. (**b**) Blow-up of artifact measure.

Then, using the measurement tool in Viscore, individual artifact measurements were taken to match those taken using traditional hand-mapping methods. These measurements were also recorded in a Microsoft Excel matrix for comparison. Once all data were recorded in a Microsoft Excel matrix, analysis of the data was conducted using functions in Excel.

## 3. Results

### 3.1. Difference between RTK and SfM Model

Table 1 displays the measurements from marker to marker calculated using both RTK and SfM model data. These data sets were used to generate the absolute value differences between RTK and SfM values displayed in the third column.

**Table 1.** This table illustrates the calculated absolute value difference between marker measurements taken by RTK and by SfM model.

| Marker to Marker | RTK Measure | SfM Measure | Absolute Value Difference |
|:---:|:---:|:---:|:---:|
| A to B | 60.398 | 59.628 | 0.770 |
| A to C | 31.799 | 31.695 | 0.104 |
| A to D | 49.979 | 49.566 | 0.413 |
| A to F | 51.125 | 50.440 | 0.685 |
| A to G | 49.187 | 48.271 | 0.916 |
| A to H | 44.581 | 44.116 | 0.465 |
| A to I | 37.162 | 37.010 | 0.152 |
| A to J | 26.515 | 26.455 | 0.060 |
| A to K | 18.524 | 18.472 | 0.052 |
| A to L | 10.772 | 10.682 | 0.090 |

**Table 1.** *Cont.*

| Marker to Marker | RTK Measure | SfM Measure | Absolute Value Difference |
|:---:|:---:|:---:|:---:|
| A to M | 15.078 | 14.919 | 0.159 |
| B to C | 38.872 | 38.990 | 0.188 |
| B to D | 23.048 | 23.242 | 0.194 |
| B to F | 9.730 | 9.817 | 0.087 |
| B to G | 17.053 | 17.100 | 0.047 |
| B to H | 15.998 | 16.081 | 0.083 |
| B to I | 25.974 | 26.143 | 0.169 |
| B to J | 36.002 | 36.090 | 0.088 |
| B to K | 41.970 | 41.776 | 0.194 |
| B to L | 50.392 | 49.930 | 0.462 |
| B to M | 47.618 | 47.286 | 0.332 |
| C to D | 20.419 | 20.444 | 0.025 |
| C to F | 32.590 | 32.612 | 0.022 |
| C to G | 36.846 | 36.699 | 0.147 |
| C to H | 27.417 | 27.472 | 0.055 |
| C to I | 19.579 | 13.627 | 0.048 |
| C to J | 10.293 | 10.264 | 0.029 |
| C to K | 21.196 | 21.210 | 0.014 |
| C to L | 26.891 | 26.840 | 0.051 |
| C to M | 17.249 | 17.056 | 0.193 |
| D to F | 21.557 | 21.642 | 0.085 |
| D to G | 29.915 | 29.900 | 0.015 |
| D to H | 20.502 | 20.564 | 0.062 |
| D to I | 14.011 | 14.015 | 0.004 |
| D to J | 23.611 | 23.612 | 0.001 |
| D to K | 34.390 | 34.308 | 0.082 |
| D to L | 42.389 | 42.165 | 0.224 |
| D to M | 35.194 | 34.943 | 0.251 |
| F to G | 9.404 | 9.360 | 0.044 |
| F to H | 6.553 | 6.538 | 0.015 |
| F to I | 19.042 | 19.105 | 0.063 |
| F to J | 27.881 | 27.883 | 0.002 |
| F to K | 32.616 | 32.374 | 0.242 |
| F to L | 10.933 | 40.485 | 0.448 |
| F to M | 38.954 | 38.617 | 0.337 |
| G to H | 10.040 | 9.957 | 0.083 |
| G to I | 23.682 | 23.626 | 0.056 |
| G to J | 29.785 | 29.628 | 0.157 |
| G to K | 31.003 | 30.566 | 0.437 |
| G to L | 29.569 | 37.885 | 0.984 |
| G to M | 39.121 | 38.574 | 0.547 |

**Table 1.** *Cont.*

| Marker to Marker | RTK Measure | SfM Measure | Absolute Value Difference |
|:---:|:---:|:---:|:---:|
| H to I | 13.994 | 14.015 | 0.071 |
| H to J | 21.686 | 21.734 | 0.048 |
| H to K | 26.081 | 25.935 | 0.146 |
| H to L | 34.436 | 34.138 | 0.298 |
| H to M | 32.492 | 32.273 | 0.219 |
| I to J | 10.810 | 10.861 | 0.051 |
| I to K | 20.535 | 20.544 | 0.009 |
| I to L | 28.753 | 28.697 | 0.056 |
| I to M | 22.906 | 22.818 | 0.088 |
| J to K | 11.726 | 11.767 | 0.041 |
| J to L | 18.968 | 18.956 | 0.012 |
| J to M | 12.156 | 12.027 | 0.129 |
| K to L | 8.607 | 8.501 | 0.106 |
| K to M | 10.004 | 9.878 | 0.126 |
| L to M | 10.705 | 10.749 | 0.044 |
| Average Difference | | | 0.174327653 |

The team found that the average absolute value difference between the distance measurements calculated using RTK data and the distance measurements calculated using the uncorrected and uncalibrated photogrammetry model was 0.174 m, with a maximum difference of 0.916 m and a minimum of 0.001 m. This average difference was calculated using absolute value because as distance is a simple measure that describes the distance between two objects. It does not matter whether the difference between the markers calculated using RTK data or Agisoft data was positive or negative, only that these values differ from one another. Additionally, a mathematical average that combines positive and negative numbers would give an artificially low value for average difference.

Appendix A is a scatter plot that illustrates the difference between the RTK calculated distances and distances calculated with the uncalibrated SfM model. The use of a scatter plot shows the large degree of agreement between the two data sets and indicates that, even though there are several seemingly large differences in measurements, this difference is negligible when the 84 m × 65 m total size of the site is taken into account.

*3.2. Statistical Analysis*

The team then conducted statistical analysis to determine a standard deviation and 95% confidence interval for the data set.

The RTK GPS data was compared to the Agisoft data to determine the three-dimensional positioning differences (errors). A total of 65 error pairs were used. Error directionality was not included in this analysis. A histogram of these errors is shown in Figure 7. The mean of these 65 errors is −12.11 cm as seen by the black vertical line in Figure 8. The differences are skewed negatively indicating that the Agisoft data is shorter than the RTK data. This may be caused by the bowling effect within the Agisoft processing. This also shows that the errors are not random with a zero mean. The standard deviation, σ, of the data is $\sigma = 21.76$ cm. The resulting 95% confidence interval is:

$$CI_{95} = (-17.4, \ -6.82) \text{ cm}.$$

This 95% confidence interval indicates a spherical error of radius $R_{95} = 5.29$ cm.

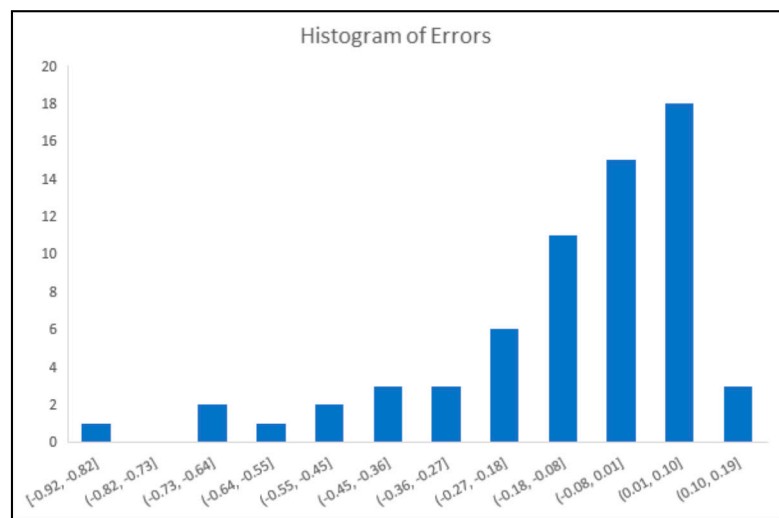

**Figure 7.** Histogram showing the distribution of the 65 error pairs.

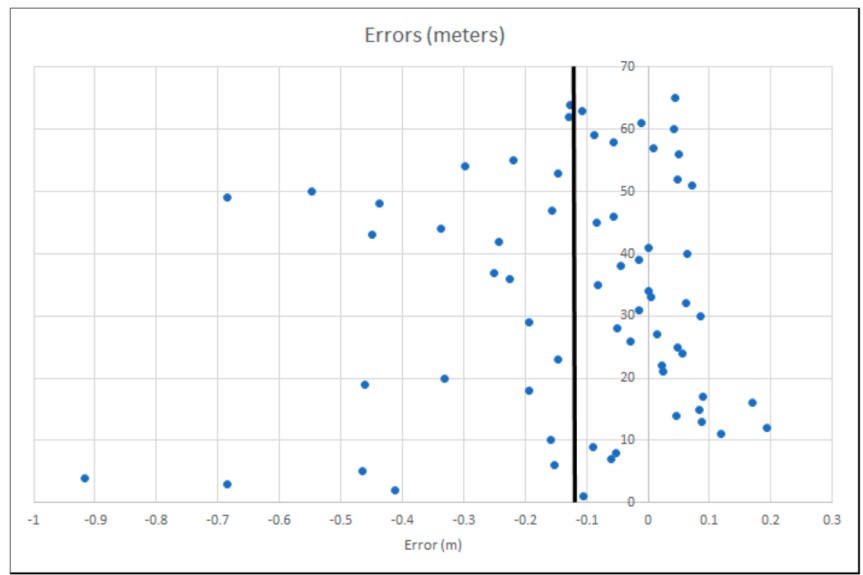

**Figure 8.** The horizontal axis shows the distribution of the 65 error pairs. The vertical line represents the mean of −12.11 cm.

### 3.3. Difference between Artifact Hand Measurements and Artifact SfM Measurements

The team measured five cement slabs, a pig iron bar, ten individual cement barrels in a line, and the line of barrels as a whole by hand on the site of the East Key Construction Wreck. These same measurements were taken in Viscore and compared in Table 2.

**Table 2.** Artifact hand measurements and artifact SfM model measurements.

|  | Meaurement 1 (cm) | | | Meaurement 2 (cm) | | | Meaurement 3 (cm) | | |
|---|---|---|---|---|---|---|---|---|---|
|  | **Hand** | **Viscore** | **Abs. Δ** | **Hand** | **Viscore** | **Abs. Δ** | **Hand** | **Viscore** | **Abs. Δ** |
| Slab #82 | 94 | 94.56 | 0.56 | 234 | 234.1 | 0.1 | — | — | — |
| Slab #83 | 166 | 164.2 | 1.8 | 40 | 39.953 | 0.05 | — | — | — |
| Barrel #1 | 67 | 66.051 | 0.949 | — | — | 0 | 67 | 66.051 | 0.949 |
| Barrel #2 | 62 | 63.197 | 1.2 | 74 | 73.703 | 0.297 | 136 | 136.9 | 0.9 |
| Barrel #3 | 65 | 64.7 | 0.3 | 141 | 140.8 | 0.2 | 206 | 205.5 | 0.5 |
| Slab #85 | 234 | 233.5 | 0.5 | 92 | 92.018 | 0.02 | — | — | — |
| Slab #86 | 255 | 252.4 | 2.6 | 42 | 41.507 | 0.49 | — | — | — |
| Slab #88 | 185 | 183.3 | 1.7 | 32 | 30.692 | 1.31 | — | — | — |

**Table 2.** *Cont.*

| | Meaurement 1 (cm) | | | Meaurement 2 (cm) | | | Meaurement 3 (cm) | | |
|---|---|---|---|---|---|---|---|---|---|
| | **Hand** | **Viscore** | **Abs. Δ** | **Hand** | **Viscore** | **Abs. Δ** | **Hand** | **Viscore** | **Abs. Δ** |
| Pig Iron Bar #90 | 10 | 10.05 | 0.05 | 10 | 9.906 | 0.09 | 73 | 72.622 | 0.38 |
| Barrel Line Length #84 | 710 | 709.7 | 0.3 | — | — | — | — | — | — |
| Ave. Abs. Difference | | | 1.072857 | | | 0.343333 | | | 0.38 |
| | **Overall Barrel Width (cm)** | | | **Barrell Origin (cm)** | | | **Barrel Terminus (cm)** | | |
| | **Hand** | **Viscore** | **Abs. Δ** | **Hand** | **Viscore** | **Abs. Δ** | **Hand** | **Viscore** | **Abs. Δ** |
| Barrel #4 | 63 | 63.9 | 0.9 | 207 | 207.1 | 0.1 | 270 | 271 | 1 |
| Barrel #5 | 69 | 65.8 | 3.2 | 278 | 281.4 | 3.4 | 347 | 347.2 | 0.2 |
| Barrel #6 | 66 | 64.3 | 1.7 | 348 | 348.8 | 0.8 | 414 | 413.1 | 0.9 |
| Barrel #7 | 67 | 65.1 | 1.9 | 430 | 430.5 | 0.5 | 497 | 495.6 | 1.4 |
| Barrel #8 | 66 | 65.4 | 0.6 | 504 | 503.9 | 0.1 | 570 | 569.3 | 0.7 |
| Barrel #9 | 65 | 66 | 1 | 573 | 571.6 | 1.4 | 638 | 637.6 | 0.4 |
| Barrel #10 | 66 | 63.3 | 2.7 | 646 | 647.4 | 1.4 | 712 | 710.7 | 1.3 |
| Ave. Abs. Difference | | | 1.714286 | | | 1.1 | | | 0.842857 |

When comparing the measurements of artifacts (the longest of which was 712 cm) taken by hand and by uncalibrated SfM model, the team found a negligible average difference of 928 cm, less than one centimeter, between the data sets.

## 4. Discussion

### 4.1. Accuracy and Precision

Balletti et al. [2] and Boyer and Lockhart [3] confirm that photogrammetry models, when properly scaled, are an accurate recording tool for submerged cultural resources on underwater archaeological sites up to 30 m in length. This study of the East Key Construction Wreck indicates that large-scale areas, even up to 5460 sq. m, can be rendered with archaeologically acceptable accuracy and precision with the use of SeaArray, as well as may be even more accurate with proper camera calibration and georectification.

Accuracy is defined as a measure of how close a value is to its true value and describes statistical bias. To test the accuracy of SeaArray photogrammetry models, the team compared measured distances generated from the RTK points and points on the SfM model. This comparison yielded a variety of errors ranging from 91.6 cm to 0.01 cm with an absolute value average error of 17.4 cm between the two data sets. The statistical analysis indicated a spherical error of radius of 5.29 cm at the 95% confidence level.

The team also compared the traditional hand measurements of artifacts and features on the site and measurements of these same objects taken in Viscore on the SfM model. These measured distances between traditional mapping techniques and measurements taken from the SfM model in Viscore varied on average by 0.928 cm. Some of this difference can be attributed to the variation between the placement of the tape measure when hand measurements were taken and the placement of the mouse when measuring in Viscore. Some level of variation between users, and even by the same user, is unavoidable, whether the measure is being derived by fiberglass tape or mouse cursor. While the team took photos of tape measure placement during the hand measuring process and attempted to place the mouse at the same points when measuring on the Viscore model, there is undeniably some amount of variation based on location. Additionally, team members in the field rounded recorded measurements to the nearest centimeter, whereas Viscore measurements were presented in the software with at least two decimal places (e.g., a hand recorded measurement of 92 cm versus a Viscore measurement of 92.01 cm). This resulted in additional differences between the traditional tape measurements and those derived from the Viscore model. We doubt that these small measurement differences are mathematically or archaeologically meaningful.

The difference in the results when measuring individual key features measured by hand or by SfM are much smaller than the difference in results when comparing marker-to-marker measurements. This partly reflects the difference between the small- and large-scales in which these measurements are made. The largest single feature measurement was 7.1 m, whereas the largest marker-to-marker measurement was 60.398 m. We predict that error in both types of measurements will be correspondingly less on sites smaller than the East Key Construction Wreck, where all measurements may be in smaller increments.

When compared to the similar underwater study conducted by Balletti et al. [2], who found photogrammetry models to be accurate up to 6 cm over a site of 36 m long when compared with RTK, a spherical error of radius of 5.29 cm at the 95% confidence level indicates similar results. This study also yields similar results to the study conducted by Uysal et al. [16], who found a digital elevation model created with aerial drone photogrammetry to be accurate to 6.62 cm from an altitude of 60 m using 30 ground control points. While this study did not find results as accurate as those found by Barry and Coakley [15], who found that aerial drone photogrammetry survey was reliable within 41 mm horizontally and 68 mm vertically with a 1 cm ground sample distance, the many variables of conducting photogrammetric survey underwater likely account for these discrepancies.

## 4.2. The Bowling Effect

The bowling effect in the uncorrected data was caused by relatively flat topography of the wreck site combined with a purposeful lack of ground control tie points and camera lens calibration. The large area of the wreck and survey area, coupled with the homogenous, low-lying topography and lack of calibrated cameras, resulted in conditions conducive to the bowling effect. Even though bowling resulted in skewed long-distance measurement data, the difference between the smaller, individual artifact measurements taken on both the corrected and uncorrected model data sets of 0.928 cm is negligible. This indicates that, had the uncalibrated, ungeorectified 3D model not fallen victim to the bowling effect, the difference in measurements between the RTK and SfM data sets would have been much smaller.

Figure 9 illustrates profile views of both the SfM model discussed in this paper and a model created with the same data set that was corrected through the integration of RTK ground control points (not discussed in this paper). It is apparent from the images that the bowling effect is responsible for some error in distance measurements from marker to marker.

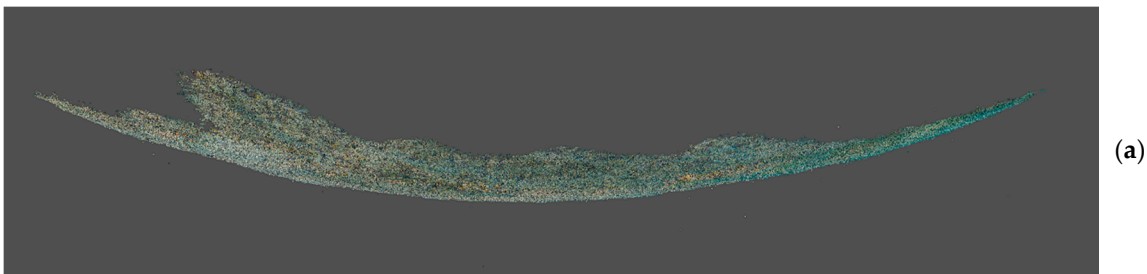

(a)

**Figure 9.** *Cont.*

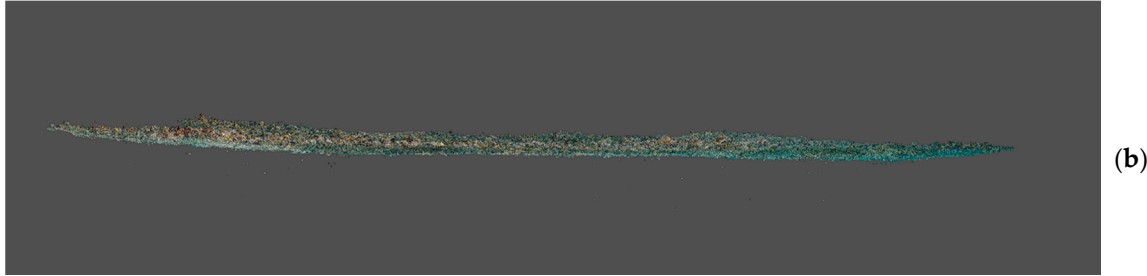

(b)

**Figure 9.** Profile views of bowled uncalibrated model (**a**) versus model corrected through ground control process (**b**).

The purpose of this study was to test the "worst case scenario" for 3D models created with SeaArray and to gain a sense of how much work was needed on a site to capture an acceptably accurate SfM model. In some ways, the presence of the bowling effect helped put this to the test by creating an even less than ideal scenario than initially imagined by the team for testing the accuracy and precision of measurements taken with a SeaArray model.

### 4.3. Photogrammetry Models vs. Hand Mapping

Holt [21] conducted a study assessing the accuracy and precision of underwater archaeological survey using traditional tape measurements of up to 20 m. He found that three-dimensional trilateration using a tape measure resulted in a position accuracy of +/−4.0 cm. Of the 304 measurements collected as part of his study, 20% of them were in error (i.e., they exceeded the 4 cm positional accuracy taken as acceptable). This was surprising and challenges the normal assumption of accuracy for underwater surveys. Additionally, the author found that there was no correlation between the size of a measurement error and the measurement length, meaning that large errors are likely to appear as often as small ones regardless of measurement length.

If Holt's [21] value is considered an acceptable amount of error, when applied to the data collected at the East Key Construction Wreck, then an acceptable measurement variation where the longest measurement is slightly more than 60 m would be 12 cm, putting the East Key Construction Wreck with an average error of 17.4 cm outside the "acceptable" level of error. However, this is a straight linear interpolation of Holt's values and ignores the realities of trying to pull very long tape measurements underwater and does not take into account our statistical analysis. It is likely that traditional tape measurements for distances exceeding the 20 m used in Holt's test case would generate larger error values. It is unlikely that archaeologists would be able to map a site of this size by hand without making a similar level of error when such things as current, tape measure slack over distance, and objects obtruding into the path of the tape measure are taken into account.

The level of variation seen by Holt [21] also occurred during the hand-measurement process of the East Key Construction Wreck. Archaeologists measured and recorded the same line of cement barrels twice while on site, once while taking the overall length of the line of barrels, and once while measuring the distance from barrel to barrel within the line, shown in Figure 10b. The result was two different lengths. When measuring the total length, the archaeologists recorded 710 cm. In contrast, when measuring the length of the line from barrel to barrel, the team recorded a total length of 712 cm (Table 2). This indicates a variation of two centimeters between measurements and, again, points to a level of inherent uncertainty in measurements taken underwater even over relatively short distances and under almost ideal conditions.

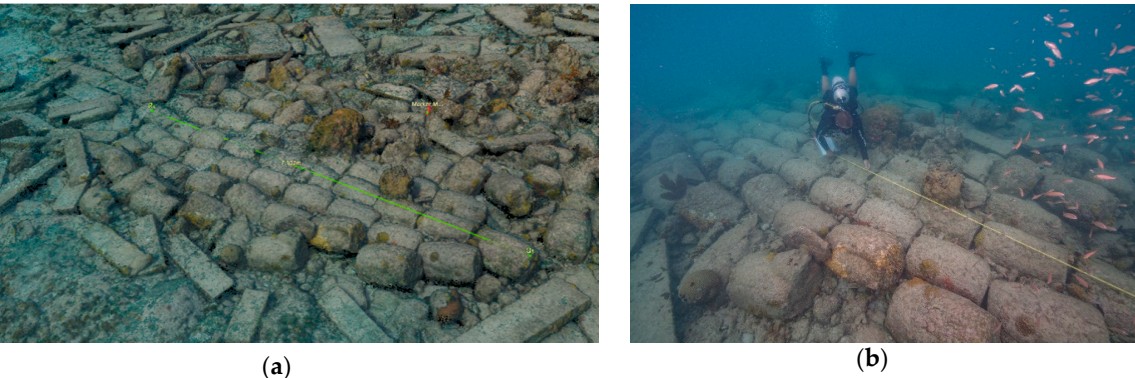

<div align="center">(<b>a</b>)　　　　　　　　　　　　　　　　　　　　　　　　　　　　　　　(<b>b</b>)</div>

**Figure 10.** Line of barrels measured as depicted in Viscore (**a**) and as measured by hand (**b**).

With the East Key Construction Wreck, we have a unique opportunity to compare traditional documentation results with SfM documentation. NPS archaeologists and students from Brown University mapped the site by hand in 1990 over a span of several weeks, resulting in a two-dimensional, traditional archaeological site plan, shown superimposed over the photogrammetry model in Figure 11 over the traditional map. This map was superimposed on the SfM model and adjusted to a best fit of features by eye—i.e., the Brown University map was moved until it had the best apparent visual overlap with the SfM model. As can be seen from Figure 11, the two maps did not cohere well. This illustrates several deficiencies in the basemap and highlights what SfM technology can contribute to traditional mapping. It is apparent from the SfM overlay that many site features are missing from the hand-drawn map, and there is disagreement between the placement of some features on the hand-drawn map and SfM model. The point of the comparison is not to denigrate the work done by the team in 1990, but rather to contrast how much the techniques and technology of SfM modeling have revolutionized documentation of underwater sites.

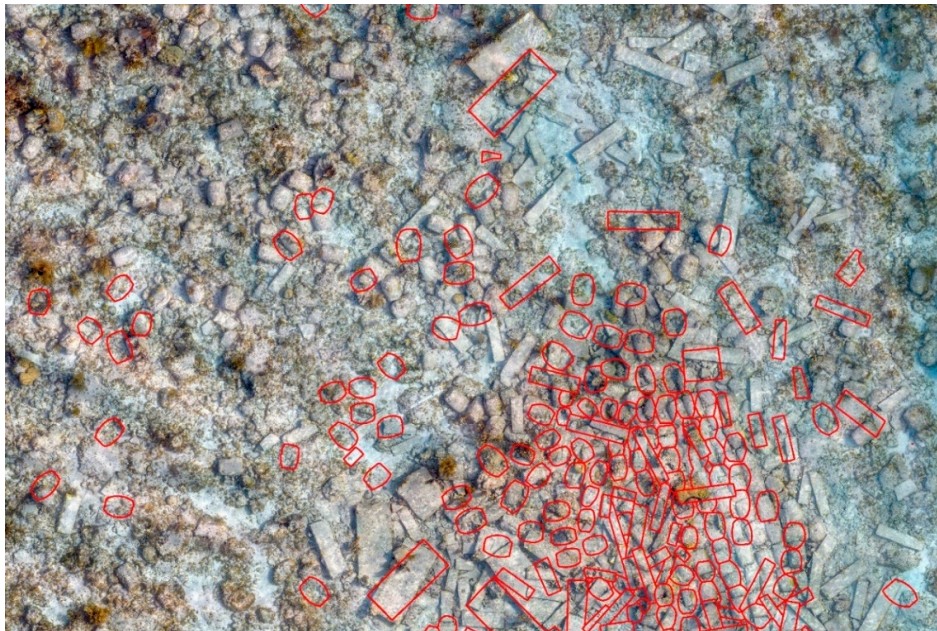

**Figure 11.** NPS hand drawn site plan from 1990 superimposed over the SfM model orthophoto.

It is, of course, understandable that the traditional map does not include many of the elements of the site, seafloor, and coral matrix in which the site is embedded. Gathering the data on which the hand-drawn map is based requires a significant amount of time and effort, and decisions about what is important enough to map and what is not are balanced against the time, staff, and other

resources available. With SeaArray, the same considerations apply, but a site the size of the East Key Construction Wreck can be recorded in a matter of hours in much greater detail than it could with weeks of hand mapping. In the span of two hours, a team of two mapped the East Key Construction Wreck. It is demonstrably faster and arguably more accurate than traditional approaches. The resulting model is visually and interpretively compelling, and it offers a tremendous amount of utility for archeologists, resource managers, and the general public. Moreover, there are secondary benefits. In terms of operations, it is cheaper and less labor intensive, in that the map data can be gathered by a smaller team with less time in the water. In terms of product and utility, the thousands of images that are collected to create a model can be used repeatedly as new questions arise, and the data can be reprocessed in future years as hardware, software, and model and rectification algorithms advance. In some very real sense, the East Key Construction Wreck site has been "digitally preserved" for future generations of managers, archeologists, and visitors to Dry Tortugas National Park.

While SfM technology promises to fundamentally change how some shipwrecks are mapped and interpreted, it entails some compromises and commitments that should be acknowledged. First and foremost is that researchers trade speed and ease of documentation for close-up detailed understanding of underwater sites. The traditional process of hand mapping and underwater drawing, while slow, imparts a tremendous amount of information to those engaged in it. Slow, careful examination of sites, features, and artifacts leads to more nuanced interpretations of site formation processes, use-life, and repairs that are keys to evidence-based scientific statements about the past. The use of SfM as a documentation tool does not replace valuable time spent underwater on a site by an archaeologist, but allows NPS SRC to better handle our mandate of protecting 1.4 million hectares of submerged land. NPS SRC has embraced SfM documentation techniques and technology for small- and large-scale baseline site documentation because of our vast mandate and relatively small team. While we acknowledge the archeological utility of hand mapping, we made a conscious decision after weighing both the pros and the cons, to move forward with SfM models as a key technology for site management and interpretation.

### 4.4. Lessons Learned and Future Research

A pitfall of SfM photogrammetry is that because the models yield such visually compelling images, it is sometimes difficult to be skeptical of the accuracy of the results they produce. The work of marrying SfM models to RTK survey points as described in this article was specifically designed to both challenge our assumptions of accuracy and provide quantifiable data about the accuracy of the models produced.

There are a number of ways in which NPS SRC is committed to improving the accuracy of SeaArray SfM models. Largely due to the appearance of the bowling effect in the East Key Construction Wreck data set and the potential for this issue to arise on other sites in the future, the team acknowledges that calibration of SeaArray's cameras is an absolutely essential step to ensuring archaeologically accurate SfM models in the future. NPS SRC is, at time of publication, designing a 4.99 m × 2.13 m camera calibration grid for SeaArray that will be deployable in the field, and plans to use this grid prior to each SeaArray project. NPS SRC has also purchased an Underwater Information Systems (UWIS) underwater GPS system that will be integrated with SeaArray's cameras to provide real-time GPS locations for SeaArray's photos.

Additionally, NPS SRC is conducting further research into the importance of EXIF data in the Agisoft Metashape alignment process, as well as lens distortion and pixel size specific to SeaArray cameras, and how this information is conveyed internally to Agisoft Metashape.

It is possible that the results produced here could be reproduced with a single camera photogrammetry system, as opposed to SeaArray's three cameras. However, use of a single camera increases the amount of time that must be spent underwater collecting data. The advantage of SeaArray's three cameras is that the system collects three times the data in the same time frame it

would take to fly a single camera system around the same site. The more overlapping images that are collected, the greater the chance of achieving successful alignment with excellent coverage.

## 5. Conclusions

This study sought to determine the accuracy of 3D models created with the SeaArray platform by comparing measurements from an uncalibrated "worst case scenario" configuration to those derived from a state of the art RTK GPS survey of the same site, as well as to evaluate the amount of time and resources necessary to capture an acceptably accurate photogrammetry model of a submerged archaeological site. Through statistical analysis, the team determined that at worst-case, in its current iteration, the photogrammetry models created with SeaArray of the East Key Construction Wreck has a margin of error of 5.29 cm on a site over 84 m in length and 65 m in width. In short, based upon our statistical analysis, measurements taken from the SfM model will fall within 5.29 cm of the true value of these same measurements taken on the East Key Construction Wreck site in 95% of all cases.

The establishment of a "worst case scenario" baseline allows NPS SRC to continue research on how to create photogrammetry models with the highest levels of accuracy given our equipment configurations. As the National Park Service moves forward with the use of photogrammetry as a tool for archaeological data collection and analysis, we can now state with a quantifiable degree of certainty that 3D models produced by SeaArray are an accurate representation of what lies underwater, but we will continue to improve this methodology. Models that are properly geo-rectified and scaled with calibrated cameras are obviously the ideal, but the case examined here also indicates that, even in a worst case, with uncalibrated lenses, no ground control, and a bowling effect caused by these two factors compounded by homogenous site topography, the resulting SfM model has considerable fidelity to the actual site itself. This has important implications for the ways in which we as archaeologists do archaeology.

NPS must often conduct rapid archaeological documentation and site monitoring due to the large number of archaeological sites located in the 1.4 million hectares of submerged public lands in its care. Confidence in the archaeological accuracy of 3D models produced by SeaArray will enable NPS archaeologists to best use SfM photogrammetry to rapidly record sites in the field, while conducting much of the associated analysis at another time, from an office. This will allow for better management of NPS time and resources, and, ultimately, better management of our shared underwater heritage.

**Author Contributions:** A.E.W.—Lead author, digital data collection and analysis, data curation, writing—original draft. D.L.C.—Project conceptualization, data collection, manuscript revision and editing, preparation of ArcGIS maps. S.M.S.—Statistical analysis, review. All authors have read and agreed to the published version of the manuscript.

**Funding:** This research was funded by the Cultural Resources Projects and Programs fund of the National Park Service.

**Acknowledgments:** The data that was used in this article were gathered by a very talented team of NPS archeologists, surveyors, and photographers. SRC Deputy Chief Brett Seymour operated SeaArray and ran numerous iterations of Agisoft/Viscore processing to create the digital models used in the analysis. NPS GPS coordinator Tim Smith designed the RTK survey, configured and installed both the base station instrumentation and the rover units, and gathered and post-processed the resulting data. Bryce Sprecher of the University of Wisconsin has developed several Metashape Professional processing workflows that allowed the team to process the large SeaArray image data efficiently. NPS Archeologist Joshua Marano assisted with all field operations and RTK work. SRC Photographer Susanna Pershern documented the RTK survey process as it occurred. The authors would also like to acknowledge and thank the staff and senior management of Dry Tortugas National Park and Everglades National Park for supporting this work. In addition, the SRC recognizes our photogrammetry partners. Marine Imaging Technologies led in the engineering, fabrication, and continued technical support of SeaArray. University of California San Diego's Cultural Heritage Engineering Initiative has proved the analytical and visualization software Viscore.

**Conflicts of Interest:** The authors declare no conflict of interest. The funders had no role in the design of the study; in the collection, analyses, or interpretation of data; in the writing of the manuscript, nor in the decision to publish the results.

## Appendix A

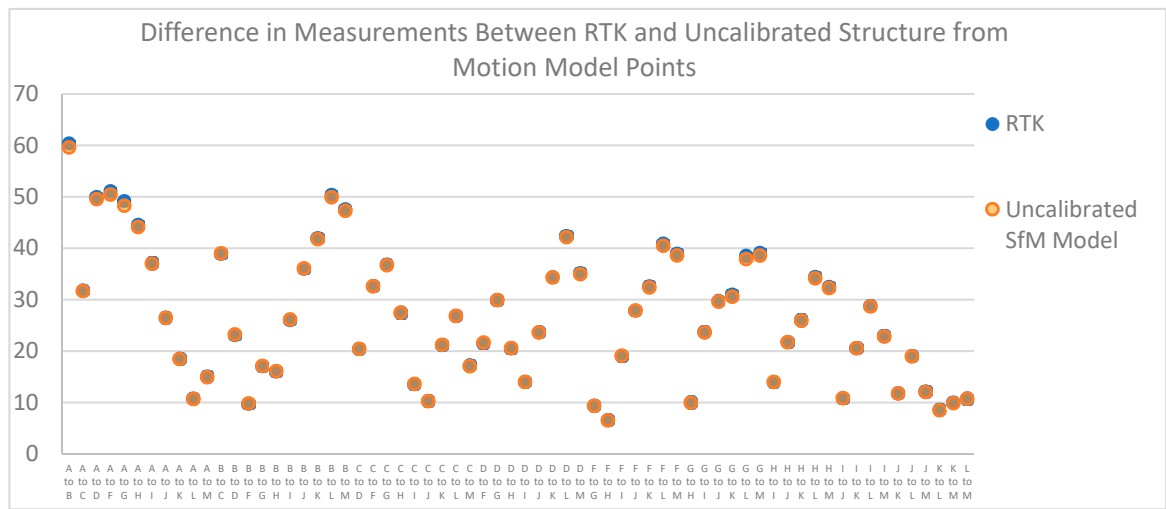

**Figure A1.** This scatter plot illustrates the difference between the RTK calculated distances and distances calculated with the uncalibrated SfM model. Distances are in meters.

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
