# Peer review of "Assessing the Accuracy of Underwater Photogrammetry for Archaeology: A Comparison of Structure from Motion Photogrammetry and Real Time Kinematic Survey at the East Key Construction Wreck"

_jmse, doi:10.3390/jmse8110849_

Round 1
Reviewer 1 Report
The article presents very interesting research very clearly.
In my humble opinion, it only lacks a short description of three key parameters determining the accuracy of photogrammetric measurements, i.e. overlap, exterior orientation parameters (in particular, the shooting distance - which determines the Ground Sampling Distance - GSD) and the forward intersection angle in optical media air - (glass) - water.
Editing notes:
- The scanned paper navigation map presented in Fig. 2 is illegible. Maybe it is better to use the electronic navigation map image available at https://nauticalcharts.noaa.gov/charts/noaa-enc.html as a background (deleting redundant information).
- The drawing in Appendix A is too large.
Author Response
Thank you for your kind remarks regarding our research. We have addressed your request for a discussion of the three key parameters in section 2.3 SeaArray Methodology. We felt that this was the best place for it (as opposed to the introduction section) as this section is the bulk of our discussion on SeaArray and how it creates photogrammetry models.
We have also updated the map with an electronic NOAA chart, as requested.
Finally, we resized the graphic in Appendix A as much as possible. I believe we are unable to make it any smaller, as this will make the data difficult to read.
Again, thank you for your kind remarks!
Reviewer 2 Report
The paper JMSE - 95346 entitled "Structure from Motion Photogrammetry and Real 4 Time Kinematic Survey at the East Key Construction 5 Wreck" presents very interesting scientific and methodological aspects. Underwater photogrammetry and Structure From Motion are now mature methodologies for the mapping of archaeological sites. RTK "Real time kinematic" surveys are essential to be able to constrain 3D models in order to obtain high accuracy in terms of metric (X, Y and Z) and geographical scale. The use of RTK is difficult to apply in the marine environment, so the experimentation conducted by the authors of the JMSE manuscript - 95346 is very original. The article deserves to be published, however, improvements are needed in the introductory part, the reference context and the state of the art of the methodologies used for underwater archaeology need to be expanded and better argued. Delete the description of the historical context as it is a scientific article. It is necessary to widen and increase the bibliography and the references on the techniques in use in the archaeological field, both traditional and innovative, in order to highlight the innovations brought by the experimentation proposed in the manuscript JMSE - 95346. The paragraph on materials and methods is very extensive and should be summarised and made more concise. For example, describe concisely the area of investigation where the experimentation was conducted, describe concisely the methods adopted to conduct the experimentation. It is necessary to optimise the editing of the figures, trying in some cases to merge them. For example, Figure 1 and Figure 2 could be in the same structure; Figure 5, Figure 6, Figure 7 and Figure 8 could also be together next to each other. Figure 9 and 10 could also be next to each other, considering that Figure 10 represents an enlargement. The formatting of the tables should be reviewed in the results paragraph. Also the paragraph of the discussions requires a careful revision, it is necessary to better compare the experimented methodology and its results with what has already been produced and present in the literature. Many of the contents of the discussion can be transferred into the results. In synthesis, the manuscript contains very interesting elements, the RTK experimentation is certainly very innovative and requires to align the writing of the article to a more scientific and descriptive standard. Review the general formatting of the manuscript according to the indications proposed by the journal.
Author Response
Thank you for your kind remarks regarding the originality and innovativeness of our research. We shortened and made more concise our discussion of the site’s historical context, but feel that it should not be deleted entirely due to this version of the journal being a special archaeology issue. We adjusted the focus of this section to be more of a description on the site’s location, and provide a short description of some of the artifacts that appear in the manuscript’s figures.
We added to the bibliography of work surrounding photogrammetry in underwater archaeology. While the breadth of research on under photogrammetry is wide, we originally kept this short because we wanted to mention only sources that were directly comparable to this manuscript (photogrammetry compared with RTK), of which there are very few. If you have additional citations that relate, please pass them along.
Regarding your comment on the lengthiness of the Materials and Methods section of the paper, we shortened descriptions where we deemed this feasible without losing pertinent information. The bulk of the length of this section is a description of our SeaArray photogrammetry platform, which we feel is necessary to describe in detail as this is the first publication in which we discuss SeaArray.
We addressed your comments regarding the grouping of Figures, and agree that this is a much better use of space within the manuscript. Additionally, we removed Table 1 which contained much of the same information as in Table 2, and greatly simplified Table 3 (now Table 2).
Finally, regarding your comment on moving elements of the discussion to the results section, we feel that the purpose of the results section is to only convey concrete numbers and findings, of which the material found in the discussion is not. We considered moving the section on the bowling effect to the results, but since much of what is discussed is hypothetical, we decided to leave it in place.
Again, thank you for your helpful comments and analysis of our work.
Round 2
Reviewer 2 Report
Manuscript jmse-953436 has been significantly improved. The article is ready to be published, congratulations to the authors for their work. However, it is requested to review and edit the figures and tables according to the indications given to the authors. Modify the text shown in Figure 9 as it is poorly readable. The figure in Appendix A could improve the representation of the results, so it could be moved.
Author Response
Thank you for your kind remarks regarding our updated manuscript. I removed the text from Figure 9, as it was impossible to show the measurements in a font large enough to see while still fitting the figure on the page. Even without, it still fulfills the purpose of the figures and shows profile views of both versions of the model. I also decreased the size of Appendix A to fit better on the page. Thank you!